# Patient Safety Comparison of Frameless and Frame-Based Stereotactic Navigation for Brain Biopsy—A Single Center Cohort Study

**DOI:** 10.3390/brainsci12091178

**Published:** 2022-09-01

**Authors:** Martin Vychopen, Johannes Wach, Valeri Borger, Matthias Schneider, Lars Eichhorn, Jaroslaw Maciaczyk, Gregor Bara, Hartmut Vatter, Mohammed Banat, Motaz Hamed

**Affiliations:** 1Department of Neurosurgery, University Hospital Bonn, 53127 Bonn, Germany; 2Department of Anesthesiology and Intensive Care Medicine, University Hospital Bonn, 53127 Bonn, Germany

**Keywords:** stereotactic biopsy, VarioGuide, mass lesion, neurooncology

## Abstract

Leksell stereotactic system-based aspiration biopsy is a common procedure in the neurosurgical treatment of deep-seated or multiple brain lesions. This study aimed to evaluate the benefit of frameless biopsy using VarioGuide compared to frame-based biopsy using the Leksell stereotactic system (LSS). We analyzed all brain biopsies using VarioGuide or LSS at our neurooncological Department of Neurosurgery in the University Hospital of Bonn between January 2018 and August 2020. We analyzed demographic data, duration of surgery, size of lesion, localization, and early complications. Uni-variable analyses were carried out on data from both groups. In total, 109 biopsies were compared (40 VarioGuide vs. 69 LSS). Patients with VarioGuide were significant older (74 (62–80) years vs. 67 (57–76) years; *p* = 0.03) and had a shorter duration of general anesthesia (163 (138–194) min vs. 193 (167–215) min, *p* < 0.001). We found no significant differences in surgery duration (VarioGuide median 28 min (IQR 20–38); LSS: median 30 min (IQR 25–39); *p* = 0.1352) or in early complication rates (5% vs. 7%; *p* = 0.644). A slightly higher false negative biopsy rate was registered in the LSS group (3 vs. 1; *p* = 0.1347). The size of the lesions also did not differ significantly between the two groups (18.31 ± 26.35 cm^3^ vs. 12.63 ± 14.62; *p* = 0.15). Our data showed that biopsies performed using VarioGuide took significantly less time than LSS biopsies and did not differ in complication rates. Both systems offered a high degree of patient safety.

## 1. Introduction

Aspiration biopsy using the Leksell stereotactic system (Elektra, Sweden) (LSS) is an established, minimally invasive neurosurgical procedure [1]. Developed by Lars Leksell in 1949 [2], the LSS has been used in many neurosurgical areas and stereotactical procedures, including placement of stereoelectroenecephalography electrodes [3] or tumor biopsies. Due to its high accuracy, LSS (Electra, Sweden) is preferred for surgical procedures in highly eloquent areas and suboccipitaly localized lesions, such as brainstem and cerebellar pathologies [4]. Since the development of neurosurgical navigation, the frameless systems have become an alternative to classic LSS for biopsies [5,6], aiming to simplify trajectory planning and shorten the procedure [7]. Many frameless systems have been introduced to the market since then [8]. VarioGuide (BrainLAB AG, Feldkirchen, Germany) aims to shorten the pre- and intraoperative part of the procedure by omitting the frame placement and necessity of CT-frame registration, which is replaced by a calibrated instrument and image-guided registration system [9]. This study compares the use of the VarioGuide (BrainLAB AG, Feldkirchen, Germany) and LSS systems for aspiration biopsy at our center.

## 2. Materials and Methods

### 2.1. Patient Selection and Inclusion Criteria

This retrospective, single center cohort study included all patients who underwent brain biopsy between January 2018 and August 2020. All patients were over 18 years of age. Clinical information about the patients was registered and documented, including age, sex, primary disease, location of biopsy, surgical procedure, surgery-related complications, and early in-hospital complications. We excluded all patients who were not classified as operable and those without complete data or follow-up information.

Early postoperative complications were assessed using a publicly available list of adverse events introduced by the Agency for Healthcare Research and Quality and the Center for Medicare and Medicaid Services and referred to as patient safety indicators (PSIs) and hospital-acquired conditions (HACs) [10,11,12]. PSIs included acute myocardial infarction, pressure ulcers, iatrogenic pneumothorax, transfusion reactions, peri- and postoperative hemorrhage, pulmonary embolism, acute postoperative respiratory failure, deep vein thrombosis, postoperative sepsis, and wound dehiscence, as well as accidental puncture or laceration. Within the group of HACs, screening was performed for pneumonia, catheter-associated urinary tract infection, surgical site infection, blood incompatibility, crushing injury, manifestation of poor glycemic control (diabetic ketoacidosis, non-ketonic hyperosmolar coma, hyperglycemic coma), fall injury, and vascular catheter-associated infection. In addition, to assess complications specific to surgery, postoperative records were screened for cerebrospinal fluid (CSF) leakage, postoperative meningitis, as well as postoperative new or worsened neurological deficits. These were classified as surgery-related complications (SSCs). Perioperative complications were defined as any postoperative adverse events with or without further surgical intervention occurring within 30 days of the initial surgery [12].

### 2.2. Indications for Surgery and Surgical Procedure

The indications for surgery were all unknown brain lesions for a histopathological diagnosis, or patients with multiple systemic tumor diseases and new brain lesions. Patients underwent one of two surgical aspiration biopsy procedures using either the VarioGuide or the LSS systems. Magnetic resonance imaging (MRI) with navigation sequencing was performed on all patients prior to surgery.

### 2.3. VagioGuide

In the VarioGuide group, image fusion and biopsy trajectory planning were also done preoperatively. Immediately after inducing the general anesthesia, the surgical team began to position the head of the patient in a Mayfield skull clamp. After that, the VarioGuide navigation system was verified (for detailed information, see Figure 1 and Figure 2). Aided by VarioGuide, a burr hole was placed at the entry point, and the biopsy specimens were obtained. Histological verification of pathological tissue was obtained before skin closure. After skin closure, the extubation was performed as soon as possible.

### 2.4. Leksell Stereotactic System

In the LSS group, periprocedural magnetic resonance imaging (MRI) with navigation sequencing was performed on all patients. CT-A was performed after securing the head in a stereotactic base frame, which required general anesthesia. The images obtained were fused with the MRI navigation sequences to plan the biopsy trajectory. The rest of the arc of the stereotactic apparatus was connected to the base frame in the operating room, and the coordinates were verified. With the exception of 2 cerebellar biopsies (performed in a semi-sitting position), all biopsies were performed in the prone position. A burr hole was placed at the calculated entry point and the biopsy specimens were obtained with the stereotactic frame. Histological verification of pathological tissue was obtained before final suturing. Subsequently, operation reports regarding duration (of surgery and general anesthesia), negative biopsy rates, and early complications (such as bleeding or infections) were analyzed. We also compared the localization and size of the mass lesions. To eliminate as much as possible bias due to the skill or experience of the surgeon, operations were carried out by only four neurosurgeons in the center; two of them treated patients with VarioGuide and the other two preferred to operate with LSS.

### 2.5. Postoperative Management

All patients received a CT scan immediately after the operation to verify the target and to rule out possible complications. Once the results of the histopathological analysis were received, all cases were reviewed by our internal neurooncological Tumor Board, consisting of neurosurgeons, radiation therapists, neurooncologists and neuroradiologists. The Board’s recommendations for further surgical treatment or for other forms of therapy—especially chemotherapy and radiotherapy in cases of tumors—were thus based on collective decision-making.

### 2.6. Statistics

Statistical analysis was performed using the computer software package SPSS (version 25, IBM Corp., Armonk, NY, USA). Fisher´s exact test and Mann–Whitney U test were used in uni-variate analysis for categorical and continuous variable. Non-parametric data were summarized by median values (first quartile–third quartile). Results with *p* < 0.05 were considered statistically significant.

## 3. Results

### 3.1. Patient’s Baseline Data

We found 109 cases where a fine needle brain biopsy was performed between 2018 and 2020. Table 1 shows their baseline data. The median age of the patients was 71 years (interquartile range (IQR) 58–78). Patients who underwent VarioGuide biopsies were significantly older (median 74 years (IQR 62–80)) than the LSS group (median 67 years (IQR 57–76); *p* = 0.0306). The ratio of male to female patients was not significant in either group. Overall, 7 patients suffered from early postoperative complications (6.4%); there was no significant difference between the VarioGuide and LSS groups. We found 3 patients with wound-healing disturbances (1 in VGB and 2 in LSS) and 4 bleeding complications (2 VGB vs. 2 LSS). The ASA score did not differ significantly between the groups (*p* = 0.1110).

### 3.2. Patient and Disease-Related Characteristics

On average, the duration of general anesthesia for patients undergoing the VarioGuide procedure was significantly shorter (median 163 min (IQR 138–194) than for those receiving the LSS procedure (median 193 min (IQR 167–215 min); *p* = 0.0009). We found no significant difference in the duration of surgery (VarioGuide: median 28 min (IQR 20–38); LSS: median 30 min (IQR 25–39); *p* = 0.1352). Table 2 shows the comparison of the outcomes between the groups. The rate of false negative biopsy and second-look surgery was slightly higher in the VarioGuide group than the LSS group (3 vs. 1; *p* = 0.1347). Overall, 3 tissue samples were taken per biopsy. We found no difference in the mean size of the lesions (VarioGuide 18.31 cm^3^ (SD ± 26.35) vs. LSS 12.63 cm^3^ (SD ± 14.62); *p* = 0.15). We also analyzed the site of the mass lesion (Table 1). The most significant localization differences were found among the brain stem (VarioGuide 0, LSS 2), cerebellum (VarioGuide 0, LSS 2) and basal ganglia lesions (VarioGuide 7 (17.5%), LSS 15 (21.7%)).

### 3.3. Histological Diagnoses

In the VGB group, we found 20 patients with glioblastoma multiforme, 11 patients with non-Hodgkin lymphoma, 4 with anaplastic astrozytoma, 1 with astrozytoma WHO Grade III, 1 with abscess and 3 false negative biopsies. In the LSS Group, we found 30 patients with glioblastoma multiforme, 20 patients with non-Hodgkin lymphoma, 7 with anaplastic astrozytoma, 7 with astrozytoma WHO Grade III, 1 with cerebral abscess, 2 with metastases of adenocarcinoma, 1 with Marchiafava Bignami disease and 1 false negative biopsy.

### 3.4. Cost Comparison

Compared to VGB, LSS showed marginally higher costs, mainly due to CT-angiography and longer anesthesia duration. However, in the German diagnoses-related group (DRG) system, the mean difference is EUR 47.56 × 30 (mean difference of anesthesia time) = EUR 83.1 for anesthesia time, resulting in a total cost difference of EUR 130.66.

## 4. Discussion

The aim of stereotactic biopsies is the identification of multiple or unknown brain lesions in order to enable optimal treatment. The two main objectives of this study were first, to evaluate the benefits of VarioGuide frameless biopsies compared to LSS frame-based biopsies at our center, and second, to define our in-house strategy for using VarioGuide or LSS for biopsies. We performed a retrospective comparison of VarioGuide frameless and LSS frame-based stereotactic biopsies at our institution. We found that the VarioGuide procedure took significantly less time under general anesthesia because there was no need for CT-A with the patient’s head secured in the stereotactic frame. As has previously been reported, some time may be spared if intraoperative imaging reduces the time needed for planning [13]. There are patient safety reasons as well as economic reasons for reducing the duration of the operation, since postoperative complications increase when the duration of the surgery is extended. A multi-center study showed that the risk of postoperative pulmonary complications was higher in patients whose surgery lasted longer than 2 h 13 min. [14] This is further underlined by a study in patients aged >75 years where a time under anesthesia of >120 min was identified as an independent risk factor for postoperative complications (odds ratio: 6.22) [15]. Although the VarioGuide procedure significantly reduced the time under anesthesia in our cohort, patient positioning should be further optimized to ensure even less time under anesthesia. Positioning of the stereotactic frame is mostly done under general anesthesia to reduce the patient’s anxiety and stress levels. Furthermore, it guarantees maximum precision and stability of the frame. For CT-A imaging, patients have to be relocated from the induction table to the CT table and back to the stretcher or the operating table after the imaging. Movements of the intubated patient can cause a number of serious adverse events, such as insufficient depth of anesthesia due to dislodgement of IV lines or serious airway problems due to accidental extubation. Physicians should avoid unnecessary movement of patients to minimize the risk of serious events. Trajectory planning, which with VarioGuide is done solely preoperatively, also shortens the time under anesthesia. VarioGuide’s markerless, image-to-patient registration technique proved itself to be the main factor for reducing cost and time [8]. Unlike CT-A time, trajectory planning time cannot be reduced in the setting with intraoperative imaging due to the need for image fusion and planning, which is performed after the imaging is done. In this study, because of its retrospective nature, we were not able to study the safety and efficacy of frameless and frame-based intracranial biopsy techniques according to the size of lesion. In a series of 270 patients, Woodworth et al. [16] reported that larger lesions, i.e., with a maximum diameter greater than 2 cm, were five times more likely to yield a diagnostic biopsy. However, in a multivariate analysis, a lesion size of less than 2 cm did not appear to be an independent predictor of a non-diagnostic biopsy. Lesion size, therefore, may not be as important an independent risk factor as commonly thought. Although we saw no statistical difference in the size of the lesions, there was a slightly higher rate of basal ganglia, brain stem or cerebellum lesions in the LSS group. The exclusive use of LSS in brain stem and cerebellum lesions in our cohort was due to previously published data of high accuracy and reasonable procedure length of stereotactic biopsies in posterior fossa pathologies [17]. Damage to blood vessels in the biopsy trajectory is often unavoidable, given the surgeon’s inability to see the result of their manipulation directly. Therefore, a distinct advantage of software used for pre-surgical planning with the frameless biopsy technique is that the surgeon is able to scroll through the trajectory path and plan to reduce the risk of hemorrhage by avoiding crossing pial and arachnoid surfaces. A bleeding diathesis will not cause a hemorrhage per se, but can turn a minor event into a serious one. Preoperative coagulation studies should therefore be routinely done before a biopsy [18]. To date, numerous studies have been trying to evaluate complication rate following brain biopsy. However, without a standardized surgical protocol, most of the results are inconsistent, reporting a wide range of outcomes (reporting death as a direct complication of the procedure vs. deaths occurring weeks or month postoperatively, which might be biased with patients who died due to natural history of the underlying disease) [19]. An excessively restrictive definition of death following brain biopsy may lead to the complication rate being underestimated, whereas an overly broad definition may lead to it being overestimated [20,21]. In clinical practice, hemorrhage is the main side effect following a brain biopsy [22]. In addition to clinical outcome, the evaluation of hemorrhage is problematic due to the definition (CT scan visible hemorrhage [20] vs. symptomatic bleeding [19]). Considering the diversity of surgical protocols and habitual practices among different institutions, a reliable comparison between the series and centers seems to be impossible. The use of a standard complication scale would be useful for precisely comparing and analyzing post-biopsy complications. Consistent with what has previously been presented in the literature [16], we saw no statistically significant difference in complication rates between the two systems. Our general rate of complications (5% for VarioGuide and 7% for LSS) is in accordance with other reported rates [8,23]. The frame-based biopsy systems are more accurate than the frameless systems [8]. Our data set was consistent with this finding, with slightly higher false negative biopsy rates for VarioGuide. The proposed 10 mm diameter margin for frameless biopsy [24] seems to us an appropriate guide for VarioGuide use. We used the fine needle aspiration biopsy technique in our cohort. Alternatively, biopsies performed with forceps with a smaller diameter than a needle biopsy could further reduce the risk of bleeding complications in frame-based biopsies. According to our institutional policy, all biopsies were performed under general anesthesia. However, due to simplification of the procedure and omitting of the CT-A in trajectory planning, neurosurgeons performing biopsies in local anesthesia might also profit from the advantages of the VarioGuide system.

### 4.1. Literature Comparison

In contemporary literature, many authors already performed evaluation of VarioGuided biopsies. Our study showed, in accordance with other publications, VarioGuide seems to be an acceptable alternative to LSS, providing reasonable safety and simplifying the procedure. The following table Table 3 summarizes the findings of comparative studies of VarioGuide system since 2000.

### 4.2. Cost Comparison

In Germany, the DRG system is used to calculate the costs of the procedure. The cost of the procedure is determined according to the average expenses of the operation determined for the treatment in the previous year. Therefore, DRG is not used for cost-recording and pricing according to the actual commercial costs. In conclusion, our comparison of cost is only valid in Germany and will differ strongly in comparison with other systems [27].

### 4.3. Conclusions

Our data showed that biopsies performed using VarioGuide took significantly less time than LSS biopsies and did not differ in complication rates. Both systems offered a high degree of patient safety. Table 4 summarises the pros and cons of both systems.

### 4.4. Limitations

This study has several limitations. Acquisition of data was retrospective, which comes with all the known and well-described kinds of bias. Patients were not randomized but treated according to the expert opinion of their neurosurgeon. Moreover, the small number of patients limits the results. The strong limitation of our data set is the fact that no biopsies were performed under local anesthesia (LA). This is mainly due to the patient collective, where many patients showed incompliance based on poor preoperative clinical conditions and were not suitable for LA biopsy. Therefore, our institutional standard is biopsy under general anesthesia. No comparison was made as to whether the biopsy was diagnostic. Our cost comparison is only valid in the DRG system. Based on the DRG system, we only provide a theoretical cost comparison and are not able to provide concrete data for this cohort.

## Figures and Tables

**Figure 1 brainsci-12-01178-f001:**
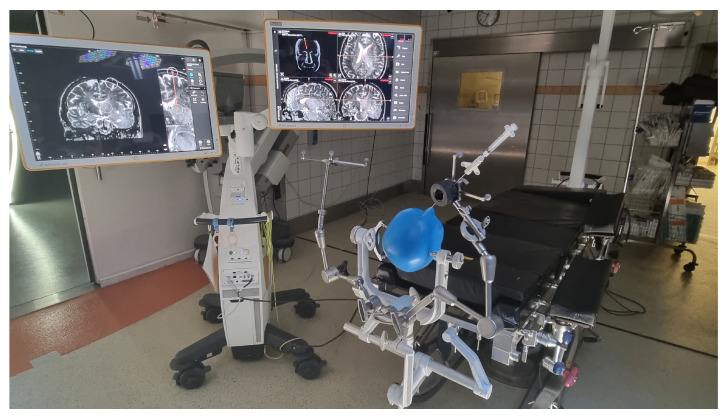
VarioGuide intraoperative setting a.

**Figure 2 brainsci-12-01178-f002:**
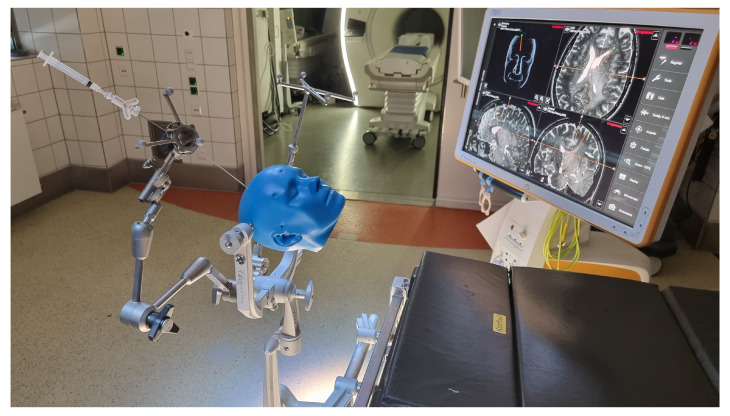
VarioGuide intraoperative setting b.

**Table 1 brainsci-12-01178-t001:** Patient characteristics.

	VarioGuide	LSS	*p* Values
Age, years, (q1–q3)	74 (62–80)	67 (57–76)	0.030
Gender, F:M	17:23	34:35	0.467
Primary tumor site			
Frontal	10	24	n.s.
Parietal	7	18	n.s.
Temporal	8	8	n.s.
Occipital	6	0	n.s.
Basal ganglia	7	15	n.s.
Brain stem	0	2	n.s.
Cerebellum	0	2	n.s.
Intraventicular	2	0	n.s.
Site of the disease			
Left	20	32	n.s.
Right	14	24	n.s.
Size of the lesion, cm^3^ (mean ± SD)	18.31 (±26.35)	12.63 (±14.62)	0.15
KPS (q1–q3)	80 (70–90)	90 (80–90)	0.08 *
ASA score ≥ 3	21	25	0.097
Preoperative neurological deficit	17	4	<0.001 *

ASA, American Society of Anesthesiology physical status classification system; q1–q3: First quartile–third quartile; KPS: Karnofsky Performance Scale; LSS: Leksell Stereotactic System; SD: standard deviation, * statistically significant, n.s: not significant.

**Table 2 brainsci-12-01178-t002:** Outcome comparison.

	VarioGuide	LSS	*p* Values
False negative biopsy rate	3	1	0.13
Surgery duration, min (q1–q3)	28 (20–38)	30 (IQR 25–39)	0.1352
Anesthesia duration, min (q1–q3)	163 (138–194)	193 (167–215)	<0.001 *
Postoperative complications	3	4	n.s.

q1–q3: First quartile–third quartile; LSS: Leksell Stereotactic System; * statistically significant, n.s: not significant.

**Table 3 brainsci-12-01178-t003:** VarioGuide comparative studies.

Author (Year)	Number of Patients	VarioGuide Evaluation
Vychopen et al. (2022)	109	Safe alternative to LSS
Ringel et al. (2009) [9]	27	100% Target localisation
Bradac et al. (2017) [25]	53	Comparable reliability to LSS
Sciortino et al. (2019) [26]	140	Comparable reliability to LSS

**Table 4 brainsci-12-01178-t004:** VarioGuide vs. LSS Summary.

	VarioGuide	Leksell Stereotactic System
Pros	Time-sparing procedure No need of pre-op CTA	More accurate, exclusive use in brainstem biopsies
Cons	Less accurate in eloquent lesions (cerebellum)	Time consuming CT-A is necessary for trajectory verification

## Data Availability

All data generated or analyzed during this study are included in this published article.

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
