# Peer review of "Patient Safety Comparison of Frameless and Frame-Based Stereotactic Navigation for Brain Biopsy—A Single Center Cohort Study"

_brainsci, 2022, doi:10.3390/brainsci12091178_

Round 1

Reviewer 1 Report

This study addresses the safety of frameless stereotactic biopsy compared with frame-based biopsy. According to the authors, the frameless biopsy took less time than frame-based biopsy and did not differ in terms of complication rates.

Please reconsider the following points.  

1.     The term “MRT” should be changed to magnetic resonance imaging (MRI).

2.     The authors should describe the precise information of specific devices such as Leksell stereotactic system and VarioGuide. For example, VarioGuide (BrainLab AG, Feldkirchen, Germany).

3.     The authors stated that the fixation of the head frame needed general anesthesia. It is not true. In most cases, we fix the frame under local anesthesia using mild sedative agents.

4.     Table 1 should be divided into two tables, namely the table indicating basic characteristics and that showing the outcomes.

5.     If the authors show the image of VarioGuide system, it would be easier to be understood.

6.     The safety and efficacy of VarioGuide were already reported in the other paper (Bradac O, Steklacova A, Nebrenska K, Vrana J, de Lacy P, Benes V: Accuracy of VarioGuide Frameless Stereotactic System Against Frame-Based Stereotaxy: Prospective, Randomized, Single-Center Study. World Neurosurg 104: 831-840, 2017). 

Author Response

Dear Editors,

Dear Reviewer,

We again appreciate the time and effort you and reviewer have dedicated to providing insightful feedback on ways to further strengthen our paper in the second round of the revision. We also hope that our edits and the responses we provide below satisfactorily address all the issues and concerns you and the reviewers have noted.

To facilitate your review of our revisions, the following is a point-by-point response to the questions and comments:

Reviewer 1

This study addresses the safety of frameless stereotactic biopsy compared with frame-based biopsy. According to the authors, the frameless biopsy took less time than frame-based biopsy and did not differ in terms of complication rates.

Please reconsider the following points.  

  1. The term “MRT” should be changed to magnetic resonance imaging (MRI).

We change the term to magnetic resonance imaging.

  1. The authors should describe the precise information of specific devices such as Leksell stereotactic system and VarioGuide. For example, VarioGuide (BrainLab AG, Feldkirchen, Germany).

Thank you for this comment. Indeed, a detailed description of the system should be added. We added more precise description in introduction part of our manuscript as follows:

Leksell stereotactic system:

Developed by Lars Leksell in 1949 \cite{24}, the LSS has been used in many neurosurgical areas and stereotactical procedures, including placement of stereoelectroenecephalography electrodes \cite{25} or tumor biopsies. Due to its high accuracy,  LSS (Electra, Sweden) is beeing prefered for surgical procederes in higly eloquent ares and suboccipitaly localized lesions such as brain-stem and crebellar pathologies.\cite{23}

VarioGuided System:

Since the development of neurosurgical navigation, the frameless systems have become an alternative to classic LSS for biopsies \cite{2, 3}, aiming to simplify trajectory planning and shorten the procedure \cite{4}. Many frameless systems have been introduced to the market since then \cite{6}. VarioGuide (BrainLAB AG, Feldkirchen, Germany) aims to shorten the pre and intraoperative part of the procedure by ommiting the frame placement and necessity of CT-frame registration, which is replaced by calibrated instrument and image guided registration system \cite{26}.This study compares the use of the VarioGuide (BrainLAB AG, Feldkirchen, Germany) and LSS systems for aspiration biopsy at our center.

Subsequently, following references were added:

  1. Nakagawa JM, The stereotactic suboccipitaltranscerebellar approach to lesions of the brainstem and the cerebellum. {\em Clin Neurol Neurosurg.} {\bf 2018}, {\em Mar}, Mar;166:10-15.
    24. Lunsford LD, Lars Leksell. Notes at the side of a raconteur. {\em Stereotact Funct Neurosurg.} {\bf 1996-1997}, {\em} ;67(3-4):153-68.
    25. Kogias E., Lars Leksell. Electrode placement for SEEG: Combining stereotactic technique with latest generation planning software for intraoperative visualization and postoperative evaluation of accuracy and accuracy predictors. {\em Clin Neurol Neurosurg.} {\bf 2022}, {\em Feb} 213:107137.
    26. Ringel F., VarioGuide: a new frameless image-guided stereotactic system--accuracy study and clinical assessment. {\em Neurosurgery.} {\bf 2009}, {\em May} 64(5 Suppl 2):365-71; discussion 371-3.

  1. The authors stated that the fixation of the head frame needed general anesthesia. It is not true. In most cases, we fix the frame under local anesthesia using mild sedative agents.

This is very relevant point we missed, which needs further explanation. As an oncological centre with overregional patient recruitment, we often perform biopsies by patients, who are not suitable for biopsy in local anethesia due to low compliance (Karnofski Index 70 and lower). Because of this, the established standard of our institution is general anesthesia. This is indeed a strong limitation of our dataset and needs to be further adressed in our limitations. However, this polymorbid patients are exactly those, who profit from VarioGuided biopsy due to simplicity of procedure and reduction of anesthesia-time.

We added a whole limitation paragraph to our manuscript:

The strong limitation of our dataset is the fact that no biopsies were performed in local anesthesia (LA). This is mainly due to the patient collective, where lot of patient show incompliance based on poor preoperative clinical condition and do not suit for the LA biopsy. Therefore, our institutional standard is biopsy in general anesthesia.

  1. Table 1 should be divided into two tables, namely the table indicating basic characteristics and that showing the outcomes.

Thank you for the comment. We see the point of dividing the Table in 2 separate ones, therefore, we added Table 2: Outcome comparison.

  1. If the authors show the image of VarioGuide system, it would be easier to be understood.

Thank you for the comment. We added a Vario-guide system Image on a specimen (plastic skull).

  1. The safety and efficacy of VarioGuide were already reported in the other paper (Bradac O, Steklacova A, Nebrenska K, Vrana J, de Lacy P, Benes V: Accuracy of VarioGuide Frameless Stereotactic System Against Frame-Based Stereotaxy: Prospective, Randomized, Single-Center Study. World Neurosurg 104: 831-840, 2017). 

Indeed, this is not the first publication evaluating the VarioGuided biopsies. However, due to the large dataset, we still aimed to perform our single center comparison.

To add the comparison with other authors, we added following paragraph and table in the discussion part:

\subsection{Literature comparison}

In contemporary literature, many authors already performed evaluation of VarioGuided biopsies. Our study showed, in accordance with other publications, VarioGuide seems to be an acceptable alternative to LSS providing reasonable safety and simplifying the procedure. The following table summarizes the findings of comparative studies of VarioGuide system since 2000.

Author´s name                                           number of patients      result of the comparison

Vychopen et al. (2022)                             & 109                                & Safe alternative to LSS         

Ringel et al. (2009) \cite{26}                  & 27                                  & 100\% Target localisation\\

Bradac et al. (2017) \cite{27}                & 53                                  & Comparable reliability to LSS\\

Sciortino et al. (2019) \cite{28}            & 140                                & Comparable reliability to LSS\\

References:

  1. Ringel F., VarioGuide: a new frameless image-guided stereotactic system--accuracy study and clinical assessment. {\em Neurosurgery.} {\bf 2009}, {\em May} 64(5 Suppl 2):365-71; discussion 371-3.
  2. Bradac O., Accuracy of VarioGuide Frameless Stereotactic System Against Frame-Based Stereotaxy: Prospective, Randomized, Single-Center Study. {\em World Neurosurgery.} {\bf 2017}, {\em Aug} 104:831-840.
  3. Sciortino T., Frameless stereotactic biopsy for precision neurosurgery: diagnostic value, safety, and accuracy. {\em Acta Neurochir (Wien).} {\bf 2019}, {\em May} 161(5):967-974.

Again, the authors wish to thank for giving us the opportunity to strengthen our manuscript. We believe that after incorporating the issues listed by the reviewer the manuscript is much clearer and has more relevance and credibility. We hope that these revisions persuade you to accept our submission.

Sincerely Yours,

Martin Vychopen

Reviewer 2 Report

The authors present a single-center cohort study comparing the patient safety of frameless (VarioGuide) versus frame-based (Leksell Stereotactic System, LSS) stereotactic navigation for brain biopsies. While knowledge of the differences between both system is valuable, comparative studies have broadly been reported:

Georgiopoulos M, Ellul J, Chroni E, Constantoyannis C. Efficacy, Safety, and Duration of a Frameless Fiducial-Less Brain Biopsy versus Frame-based Stereotactic Biopsy: A Prospective Randomized Study. J Neurol Surg A Cent Eur Neurosurg. 2018;79(1):31-38. doi:10.1055/s-0037-1602697

Kesserwan MA, Shakil H, Lannon M, et al. Frame-based versus frameless stereotactic brain biopsies: A systematic review and meta-analysis. Surg Neurol Int. 2021;12:52. Published 2021 Feb 10. doi:10.25259/SNI_824_2020

Dammers, R., Haitsma, I., Schouten, J. et al. Safety and efficacy of frameless and frame-based intracranial biopsy techniques. Acta Neurochir (Wien) 150, 23 (2008). https://doi.org/10.1007/s00701-007-1473-x

Dhawan S, He Y, Bartek J Jr, Alattar AA, Chen CC. Comparison of Frame-Based Versus Frameless Intracranial Stereotactic Biopsy: Systematic Review and Meta-Analysis. World Neurosurg. 2019;127:607-616.e4. doi:10.1016/j.wneu.2019.04.016

The findings of this study are in line with results from prior studies (equal patient safety, no statistical differences in complication rates/morbidity/mortality, shorter procedural time in frameless biopsies, no difference in overall OR time). Therefore, the study provides no novel insights/findings to the current literature.

One of the main objectives of the present study were to evaluate the benefits of VarioGuide frameless biopsies compared to the LSS system. A table summarizing the benefits and disadvantages of each technique would benefit the manuscript in my opinion.

The authors report a statistically indifferent early post-interventional complication rate (5% vs 7%, p=0.644). Please name the individual complications or present them in a table.

Although this may not be directly related with evaluating patient safety, the final histological diagnoses of the lesions should be integrated.

Cost studies or studies evaluating patient preference may add novelty to this well-explored topic.

Reviewer 3 Report

The authors evaluated the benefit 2 of frameless biopsy using VarioGuide compared to frame-based biopsy using the Leksell Stereotactic 3 System (LSS). They analyzed throughly demographic data, duration of surgery, size of lesion, localization, and early complications in 109 patients who underwent brain tumor biopsy (40 VarioGuide and 69 LSS). The authors found that biopsies performed using VarioGuide took significantly less time than LSS biopsies and did not differ in complication rates. Moreover, both systems offered a high degree of patient safety.

This is an interesting study comparing different methods of biopsy of brain lesions. It provides additional insights into current knowledge of the field. The study is well designed, and the manuscript is well written.

Major issue

  1. The introduction is too short. Please describe that biopsies are also done to assess the prognostic and predictive markers in gliomas which are particularly important in the new 2021 WHO CNS5 classification. Please see: doi: 10.3390/ijms221910373 and  DOI: 10.1093/neuonc/noab106. 

  2. There is no indication in which position were patients (prone, semi-sitted, or maybe different?)

  3. How were patients divided into the VarioGuide and LSS groups? Why were the VarioGuide biopsy patients older?

  4. Why did patients undergoing LSS biopsy undergo general anesthesia? It is possible to perform this procedure under a local anesthetic. In fact, it may be related to the patient's anxiety and stress levels, but it is less burdensome for the patient. What type of anesthesia was used for the VarioGuide biopsy group?

  5. There is a recent study in which researchers described infratentorial stereotactic biopsy using a frame-based system. The authors reported high effectiveness and low complication rate, and the procedure lasted only about 40 min - see DOI: 10.3390/brainsci11111432. 

Minor issue:

  1. Please rename table 1. (e.g. Patient characteristics, or group comparison ect.)

  2. Add to limitations: no comparison was made as to whether the biopsy was diagnostic

  3. Present in tabular form disease-related characteristics (statistical significance in procedure duration)

Round 2

Reviewer 1 Report

Please modify some misspellings (ex. Elektra, cerebellar etc.).

Author Response

Thank you for your attention. We corrected all the spelling mistakes in our manuscript.

Reviewer 2 Report

Thank you for revising the manuscript and providing more insights into the study results.

Regarding the complication rate, you state that 7 patients had postoperative complications, of which 1 had a wound infection and 4 had bleeding complications. This does not sum to 7; please revise.

Since you mention cost data and the DRG system, you should elaborate on the DRG system because it may be unfamiliar to people from countries with privatized health care systems. Please also state that this is a theoretical cost calculation since you are not providing concrete data for this cohort.

Thank you.

Author Response

Thank you for the attention. We stated our numbers correctly:

We found 3 patients with wound-healing disturbance (1 in VGB and 2 in LSS) and 4 bleeding complications (2 VGB vs. 2 LSS).

Thank you for the relevant points. Indeed, the detailed cost analysis is strongly limited by DRG system, which has to be stated. We added a paragraph on brief explanation of DRG system and a limitation to our cost analysis:

\subsection{Cost comparison}

In Germany, the DRG system is used to calculate the costs. The cost of the procedure is determined according to the average expenses of the operation determined for the treatment in the previous year. Therefore, DRG is not used for cost-recording and pricing according to the actual commertial costs. In conclusion, our comparison of cost is only valid in Germany and will differ strongly in comparison with other systems. \cite{30} 

References:

\bibitem{30} Vera A., Die "Industrialisierung" des Krankenhauswesens durch DRG-Fallpauschalen--eine interdisziplinäre Analyse [The "Industrialisation" of the hospital sector by DRG-based prospective payment systems--an interdisciplinary analysis]. {\em Gesundheitswesen.} {\bf 2009}, {\em Mar} 71(3):e10-7.

Limitations:

Our cost comparison is only valid in DRG system. Based on the DRG system, we only provide theoretical cost comparison and are not able to provide concrete data for this cohort.

Reviewer 3 Report

Thank you very much for replying to my suggestions. The authors properly addressed my criticisms, and the manuscript is much improved now. However, I have one minor comment.

There is a recent study in which researchers described infratentorial stereotactic biopsy using a frame-based system. The authors reported high effectiveness and low complication rate, and the procedure lasted only about 40 min - see DOI: 10.3390/brainsci11111432. 

Please cite the study's findings as evidence of the high efficacy and low complication rate of frame-based biopsy, even in tumors with a hazardous location, such as brainstem tumors.

Author Response

Thank you for your evaluation. As you correctly stated, the frame-based biopsy was exclusively done in such tricky localisations. Therefore, we cite the paper you mentioned and added following in our discussion:

The exclusive use of LSS in brain stem and cerebellum lesions in our cohort was due to previously published data of high accuracy and reasonable procedure length of stereotactic biopsies in posterior fossa pathologies. \cite{29}

Subsequently, following reference was added:

\bibitem{29} Furtak J., Infratentorial Stereotactic Biopsy of Brainstem and Cerebellar Lesions. {\em Brain Sci.} {\bf 2021}, {\em Oct} 11(11):1432.